

# Illuminating the druggable genome through patent bioactivity data

Maria P. Magariños[1], Anna Gaulton[1,2], Eloy Félix[1], Tevfik Kiziloren[1], Ricardo Arcila[1], Tudor I. Oprea[3] and Andrew R. Leach[1]

[1] EMBL-EBI, Hinxton, United Kingdom
[2] Exscientia, Oxford, United Kingdom
[3] Translational informatics Division, Department of Internal Medicine, School of Medicine, University of New Mexico, Albuquerque, United States

## ABSTRACT

The patent literature is a potentially valuable source of bioactivity data. In this article we describe a process to prioritise 3.7 million life science relevant patents obtained from the SureChEMBL database (https://www.surechembl.org/), according to how likely they were to contain bioactivity data for potent small molecules on less-studied targets, based on the classification developed by the Illuminating the Druggable Genome (IDG) project. The overall goal was to select a smaller number of patents that could be manually curated and incorporated into the ChEMBL database. Using relatively simple annotation and filtering pipelines, we have been able to identify a substantial number of patents containing quantitative bioactivity data for understudied targets that had not previously been reported in the peer-reviewed medicinal chemistry literature. We quantify the added value of such methods in terms of the numbers of targets that are so identified, and provide some specific illustrative examples. Our work underlines the potential value in searching the patent corpus in addition to the more traditional peer-reviewed literature. The small molecules found in these patents, together with their measured activity against the targets, are now accessible *via* the ChEMBL database.

## INTRODUCTION

One of the most useful and compelling pieces of evidence for the druggability of a new biological target is the existence of molecules that bind with sufficient affinity to modulate the biological activity of the target. However, only about 11% of the proteome has either an approved drug or a compound known to modulate it (*Sheils et al., 2021*). Chemical probes represent a special type of small molecule for use in target validation studies, not only having good activity against the target but also selectivity, cellular activity, and potentially other relevant criteria (*Workman & Collins, 2010*; *Garbaccio & Parmee, 2016*) and are subjected to a peer review process to ensure the quality of any conclusions when used by the wider community.

The availability of open-access, public databases such as ChEMBL (*Mendez et al., 2019*) has greatly simplified the task of identifying potential molecules by providing easy access to

Corresponding author
Andrew R. Leach, arl@ebi.ac.uk

more than 19 million bioactivity data points on almost 2 million compounds. The source of these data is primarily the peer-reviewed scientific literature manually extracted by curators; but some of the data has been integrated from other databases including PubChem BioAssay (*Kim et al., 2021*) and BindingDB (*Gilson et al., 2016*), and data are also deposited directly from experimental groups.

An additional and potentially valuable source of information and data on bioactive molecules is the patent literature. In drug discovery, patents are routinely filed to protect novel inventions, by both industrial organisations and academic institutions. The relationship between the patent and the "traditional" (academic journal, peer-reviewed) literature has been examined in various published studies, mostly focussed on key questions relating to overlap/duplication and publication date. For example, in a 2009 article the authors found that just 6% of compounds in patents also appeared in the scientific literature in one of the commercial sources included in their study (GVKBIO) (*Southan, Varkonyi & Muresan, 2009*). A later study examined 130 drug-target pairs and on average found them published in patents 3.7 years earlier than in scientific articles (*Senger, 2017*). A 2017 study concluded that the first molecules for a novel target are more likely to be published first in the literature, whereas novel small molecules more frequently appear first in patents than in literature, regardless of which targets they modulate (*Ashenden et al., 2017*). Finally, a more recent study selected medicinal patents published between 2014 and 2019 and identified patents with information on small molecules, antibodies and vaccines that could potentially be repurposed for cancer related therapies. Some of the drug-disease links found were not present in scientific literature, while others were found in the articles; in some cases these were published before the patents and in others afterwards (*Mucke, 2021*).

These and similar studies suggest that the patent corpus potentially represents a wealth of information that is not available elsewhere and/or may appear in the scientific literature only after a significant time delay.

Previous studies that have attempted to search or annotate pharmaceutical patents with target-compound information include *Akhondi et al. (2014)* who produced a set of 198 patents manually annotated with chemical compounds, diseases, targets and modes of action by four different groups of curators; *Suriyawongkul, Southan & Muresan (2010)* who tried to identify targets in titles, abstracts and claims of patents that contained bioactive compound information, combining the search for target names with the search for some keywords that were related to bioactivity data; *Tyrchan et al. (2012)* who compared different methods to extract the key compound from a given patent, and then applied one of the methods to inform the design of AXL kinase inhibitors; *Fechete et al. (2011)* who searched full-text patents using keywords related to diabetic nephropathy and further narrowed the search by rules related to frequency and/or patent section of the keywords found, and subsequently extracted the genes mentioned in the claims section of these patents; *Gigani et al. (2016)* who performed a search in the SureChEMBL database (*Papadatos et al., 2016*) using keyword and/or chemical structure searches, with the goal of identifying patents with compounds that could activate the $BK_{Ca}$ channel; and *Gadiya et al. (2022)* who developed a tool (PEMT) to identify patents using genes as a starting

point, searching for compounds in ChEMBL with activity data for each gene and then searching SureChEMBL using the compounds found in ChEMBL. These, and similar studies, also confirm that extracting information from patents poses many challenges, given the length and complexity of these documents.

Patent data are currently freely available from a number of resources, including Google Patents (https://patents.google.com/), The Lens (https://www.lens.org/), Espacenet (https://www.epo.org/searching-for-patents/technical/espacenet.html), Patentscope (https://patentscope.wipo.int/search/en/search.jsf) and Free Patents Online (https://www.freepatentsonline.com/). All of these systems allow searching for patents using various criteria. Pubchem provides links to the Patentscope database from the World Intellectual Property Organization (WIPO) for more than 16 million compounds, which allows users to find the patents associated with each of these chemical structures (*Kim et al., 2021*). BindingDB (*Gilson et al., 2016*) includes a curated set of US granted patents, from which protein-compound activity data is extracted.

In the work reported here, we use SureChEMBL (*Papadatos et al., 2016*) (https://www.surechembl.org/), which is a fully automated, chemical-structure-enabled database providing the research community with open and free access to the patent literature. Currently, SureChEMBL sources data from both patent applications and granted patents *via* full text patents from the United States Patent and Trademark Office (USPTO), the European Patent Office (EPO) and the World Intellectual Property Organization (WIPO), and titles and abstracts from the Japanese patent office (JPO).

SureChEMBL currently contains ~140 million patents with ~50,000 added monthly. Of these, ~25 million patents are chemically annotated. Approximately 80,000 new compounds are extracted and added each month to the SureChEMBL chemistry database which currently contains more than 25 million unique structures. The pipeline for the extraction of chemical compounds from patents has been described in detail (*Papadatos et al., 2016*). In summary, chemical entity names, images and molfiles associated with each patent are converted into chemical structures and then registered into a structure-searchable database. This process is fully automated, without requiring any manual step or curation. The data in SureChEMBL can be accessed *via* a web interface that enables users to perform text and chemical structure queries, filter the output and then display the results. The complete set of chemical structures and patent associations is also available for download.

The US National Institutes of Health established the Illuminating the Druggable Genome (henceforth IDG) project in 2014 (https://commonfund.nih.gov/idg), with the goal of increasing the knowledge about understudied proteins that belong to well-studied protein families, such as ion channels, G-protein coupled receptors (GPCR) and protein kinases. One of the key deliverables of the IDG project is an informatics platform, Pharos (https://pharos.nih.gov/), that provides researchers with free access to relevant data on targets. An important aspect of the IDG project (and the data in Pharos) is the classification of human proteins into four target development level (TDL) families, based on how well studied these proteins are. In the Tclin category are targets of at least one approved drug; Tchem targets do not have approved drugs but are modulated by at least

one small molecule with a potency above the cut-off specified for the target protein family (≤30 nM for kinases, ≤100 nM for GPCRs and nuclear receptors, ≤10 μM for ion channels and ≤1 μM for other target families); Tbio targets do not have chemistry qualifying for the Tclin/Tchem categories but satisfy the criteria described at http://juniper. health.unm.edu/tcrd/; while Tdark targets are understudied proteins with little annotation (*Oprea et al., 2018*). Of particular relevance to the work here is the availability of small molecule modulators for new targets, consistent with other work suggesting that the lack of high-quality chemical probes for understudied targets is an important cause for lack of interest (*Edwards et al., 2011*; *Oprea et al., 2018*). The default IDG process uses bioactivity data from ChEMBL (*Mendez et al., 2019*), as well as from Guide to Pharmacology, which contains manually curated information on ligands and drug targets (*Armstrong et al., 2020*), and DrugCentral (*Avram et al., 2021*), which contains bioactivity data annotated from a variety of sources, including the scientific literature. DrugCentral has also information on drugs that have been approved by the United States Food and Drug Administration, the European Medicines Agency, and the Pharmaceuticals and Medical Devices Agency in Japan. The DrugCentral drug information is used in the IDG workflow to identify the targets that belong to the Tclin category (TCRD Home Page, http://juniper. health.unm.edu/tcrd/). At the time of writing, Tclin proteins represent ~3% of the human proteome; Tchem proteins represent ~8%; Tbio proteins represent ~58%; and Tdark proteins represent ~31% (*Sheils et al., 2021*).

In this article, we describe methods to systematically mine the SureChEMBL patent corpus to identify new bioactivity data for Tdark/Tbio targets, with the aim of (1) including the bioactivity data in the ChEMBL database and (2) promoting some of these targets to IDG Tchem status.

## METHODS

Patents were processed using perl scripts written for this project, accessible *via* a GitHub repository (https://github.com/chembl/idg_patents_paper).

The starting point for our work was the set of patents extracted from SureChEMBL covering the years 2012 to 2018, flagged as life-science relevant according to the International Patent Classification (IPC) (https://www.wipo.int/classifications/ipc/en/) or the Cooperative Patent Classification (CPC) codes (https://worldwide.espacenet.com/classification) present in the patents. These codes classify the patents into different areas of technology. The specific codes taken into account by the life science flag are: A01, A23, A24, A61, A62B, C05, C06, C07, C08, C09, C10, C11, C12, C13, C14, G01N, which cover a broader set of patents than required but is still useful to filter out many patents that would not be relevant. This resulted in a set of 3.7 million patents.

The goal was to find patents with bioactivity data on small molecules against understudied targets (Tdark or Tbio categories according to the IDG classification). Firstly, in order to determine which patents were likely to have bioactivity data, the files corresponding to the patents were processed to identify tables containing the following keywords: IC50, XC50, EC50, AC50, Ki, Kd, pIC50, pXC50, pEC50, pAC50, −log(IC50),

−log(XC50), −log(EC50), −log(AC50), concentration to inhibit, IC-50, XC-50, EC-50, AC-50, IC 50, XC 50, EC 50, AC 50.

Out of the 3.7 million patents, 69,289 patents (2%) were thus flagged as potentially containing bioactivity data in tables (for simplicity called "patents with bioactivity tables").

Separately, we identified patents that might contain information about IDG Tbio and Tdark targets. A list of Tdark and Tbio IDG target names and gene symbols was obtained from the Target Central Resource Database (36,044 target names/symbols) (TCRD Home Page, http://juniper.health.unm.edu/tcrd/). We searched for these target names and their gene symbols in the patent titles, abstracts, descriptions and claims sections, in the context of specific phrases that could indicate bioactivity data of small molecules against them:

- X inhibitors
- Inhibitors of X
- X inhibitor
- Modulators of X
- Modulation of X
- Targeting X
- X modulators
- Binding specifically to X
- X mutants
- Inhibit X
- Antibodies recognis|zing X
- Modulating the X
- Selective X inhibitors
- X antagonists
- X agonist
- X selective binding compounds
- Activity of X
- X antibodies
- X activity
- Inhibitor of X
- X binding
- Antibodies directed against X
- Treatment of X related
- Antibody for X
- Anti-X antibody
- Human anti-X
- Antibodies to X
- High X affinity

- Inhibiting X
- Blocks|block X
- Blocking X
- Ligand|ligands for X
- Compounds that interact with X
- Modulating the function of X
- X ligand|ligands

The combination of these two procedures allowed us to classify the patents into six groups: patents with bioactivity tables, and targets mentioned in titles or abstracts; patents with bioactivity tables, and targets mentioned in descriptions or claims sections; patents without bioactivity tables, and targets mentioned in titles or abstracts; patents without bioactivity tables, and targets mentioned in descriptions and claims; patents with bioactivity tables but no targets; and patents without bioactivity tables and without targets (Fig. 1). This was done with the goal of prioritising the patents, with the expectation that most data would be found in Group 1, followed by Group 2; we expected Group 3 and Group 4 to contain fewer patents with bioactivity data (given that they were not flagged as containing bioactivity tables). The patents in Group 5 and Group 6 did not have target matches and for this reason were not expected to have bioactivity data against the understudied targets of interest to us.

Following this automated annotation/filtering process, a number of patents from each group were manually examined to confirm the presence of the correct Tbio/Tdark target, the presence of quantitative bioactivity measurements, and that the Tbio/Tdark target was the molecular target to which these bioactivity measurements applied. This final check is required because some of the patents were found to have data only on targets that did not belong to the IDG list of understudied targets; other patents did contain data exclusively on the targets of particular interest to us. Some patents fell into both categories.

For patents with confirmed bioactivity data, details of compounds synthesised, biological assays performed, and bioactivity measurements were manually extracted according to the standard ChEMBL curation procedure described previously (Gaulton et al., 2015) and loaded into the ChEMBL database. Briefly, structures and names of all tested compounds were extracted, together with a description of the assays performed, name of the targets, species, and measurement values and units. Compound structures were standardised and integrated into ChEMBL, mapping them to an existing structure or creating a new entry in the database as appropriate.

In addition to registering the reported measurement values, the bioactivity data obtained was standardised to facilitate comparison of results for common activity types. Bioactivity data was also mapped to existing ChEMBL targets according to species and sequence or accession. When this was not possible a new target was created and then mapped to the corresponding assay.

All bioactivities against all the targets present in these patents (irrespective of their inclusion or not in the IDG Tbio/Tdark categories) were extracted by the curators.

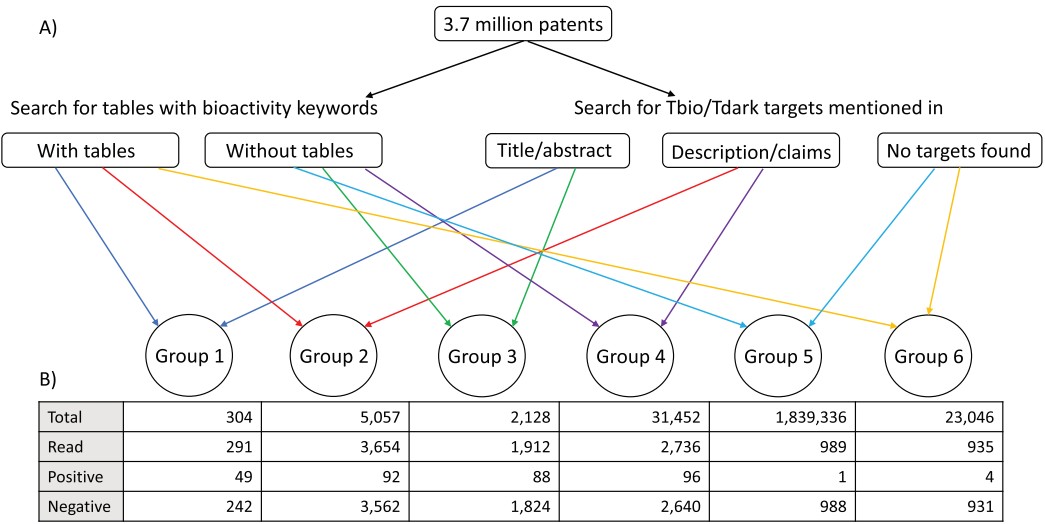

| | Group 1 | Group 2 | Group 3 | Group 4 | Group 5 | Group 6 |
|---|---|---|---|---|---|---|
| Total | 304 | 5,057 | 2,128 | 31,452 | 1,839,336 | 23,046 |
| Read | 291 | 3,654 | 1,912 | 2,736 | 989 | 935 |
| Positive | 49 | 92 | 88 | 96 | 1 | 4 |
| Negative | 242 | 3,562 | 1,824 | 2,640 | 988 | 931 |

**Figure 1  Patent group classification.** (A) A total of 3.7 million patents were scanned for the presence of (1) tables with bioactivity keywords and (2) understudied target names mentioned in the context of specific phrases in the titles, abstracts, description and claims sections of the patents. According to the result of each of these two independent processes, the patents could be classified in the following groups: Group 1: patents with bioactivity tables, and targets mentioned in titles or abstracts; Group 2: patents with bioactivity tables, and targets mentioned in descriptions or claims sections; Group 3: patents without bioactivity tables, and targets mentioned in titles or abstracts; Group 4: patents without bioactivity tables, and targets mentioned in descriptions and claims; Group 5: patents with bioactivity tables but no targets; Group 6: patents without bioactivity tables and without targets. (B) A subset of patent families in each group was manually examined. Total: total number of patent families that belonged to each group; Read: number of patent families of each group that were read to determine the presence of relevant data; Positive: number of patent families read that had bioactivity data of small molecules against at least one understudied target; Negative: number of patent families read that did not have bioactivity data of small molecules on understudied targets.

## RESULTS

As a result of this work, bioactivity data from 225 patents were loaded into ChEMBL, corresponding to 657 targets (including single proteins, protein families, protein complexes, organisms, cell lines and protein-protein interactions) and 18,319 compounds. For 145 of these targets, this represents the only source of information of bioactivity data in ChEMBL.

A patent family is a set of patents of identical content (European Patent Office, https://www.epo.org/searching-for-patents/helpful-resources/first-time-here/patent-families/docdb.html). The scripts described here were run against every patent in SureChEMBL that belongs to the set of 2012–2018 patents flagged as life science related. In order to avoid duplication of effort, patents were grouped by patent family. For this reason, the patent counts in the sections below are given as number of patent families rather than number of patents.

We examined the distribution of positive and negative patent families among the different groups shown in Fig. 1, to identify which group or groups were more or less likely to contain useful information, as this might facilitate the task of identifying the most useful patents for future analyses. The group that had the highest percentage of positive patents

was Group 1: 49 positive patent families in the 291 families examined (16.8%), followed by Group 3: 88 positive families in the 1,912 examined (4.6%). There was one patent family in common between the positive patents of these two groups. Group 2 had 92 positive patent families, but 46 of them were already present in Group 1. Group 4 had 96 positive patent families, but 86 of them were already present in Group 3. There were very few patents with data in Group 5 or Group 6 (0.1% and 0.4% patent families of the examined ones, respectively) (Fig. 1).

A full list of patents and the targets they contain can be found in Table S1. Note that one patent is omitted from this list (US-8409550-B2) because it contains data against a target from *Bos taurus*, whereas IDG is focussed solely on human targets. A total of 76 of these targets had at least one compound with bioactivity data values within the cut-off for its target family, as defined by the target class-specific IDG criteria outlined earlier. Table S2 shows which targets had bioactivity data within the cut-offs for its target family, and Table S3 shows how many patents, total compounds and compounds within the cut-off were found for each IDG target class.

As BindingDB also extracts data from patents, we were interested in examining the overlap between the two data sets. For all the targets found, we performed a search by target name in BindingDB with the goal of comparing the results from the two different databases. Because BindingDB extracts only US granted patents, we used the patent family identifier to do the comparison. We found 33 targets in both BindingDB and the dataset from our method. Of the 70 patent families found by our method for these targets, 20 were also found in BindingDB. A total of 50 families were found exclusively with our method, and 34 families were found exclusively by BindingDB. In most cases, the patents that were missed had targets mentioned using a name that was not on our list of targets to find (for example, "CH24H" instead of "Cholesterol 24-hydroxylase"). In other cases the patents belonged to Group 3 or Group 4 and were not part of the set of patents that we selected to read.

## Examples of understudied targets with bioactivities found in patents

In this section we briefly describe three specific examples of targets for which we were able to identify and curate bioactivity data from the patent workflow described above. Some of the compounds found for each target are shown in Fig. 2.

### LATS1

LATS1 is a Ser/Thr kinase that belongs to the LATS (large tumor suppressor) family (*Xu et al., 2015*). It is a component of the Hipo pathway which is involved in cancer, organ development, growth and regeneration (*Fu, Plouffe & Guan, 2017*), and cell contact inhibition (*Zheng & Pan, 2019*).

This kinase is conserved among several organisms, such as yeast, nematodes, flies, and mammals. In humans, LATS1 can be found in high levels in most tissues, and it has a role in regulation of mitosis and apoptosis. In some types of cancer there is evidence of mutations in LATS1, and of LATS1 inactivation through promoter hypermethylation in others (*Furth & Aylon, 2017*).

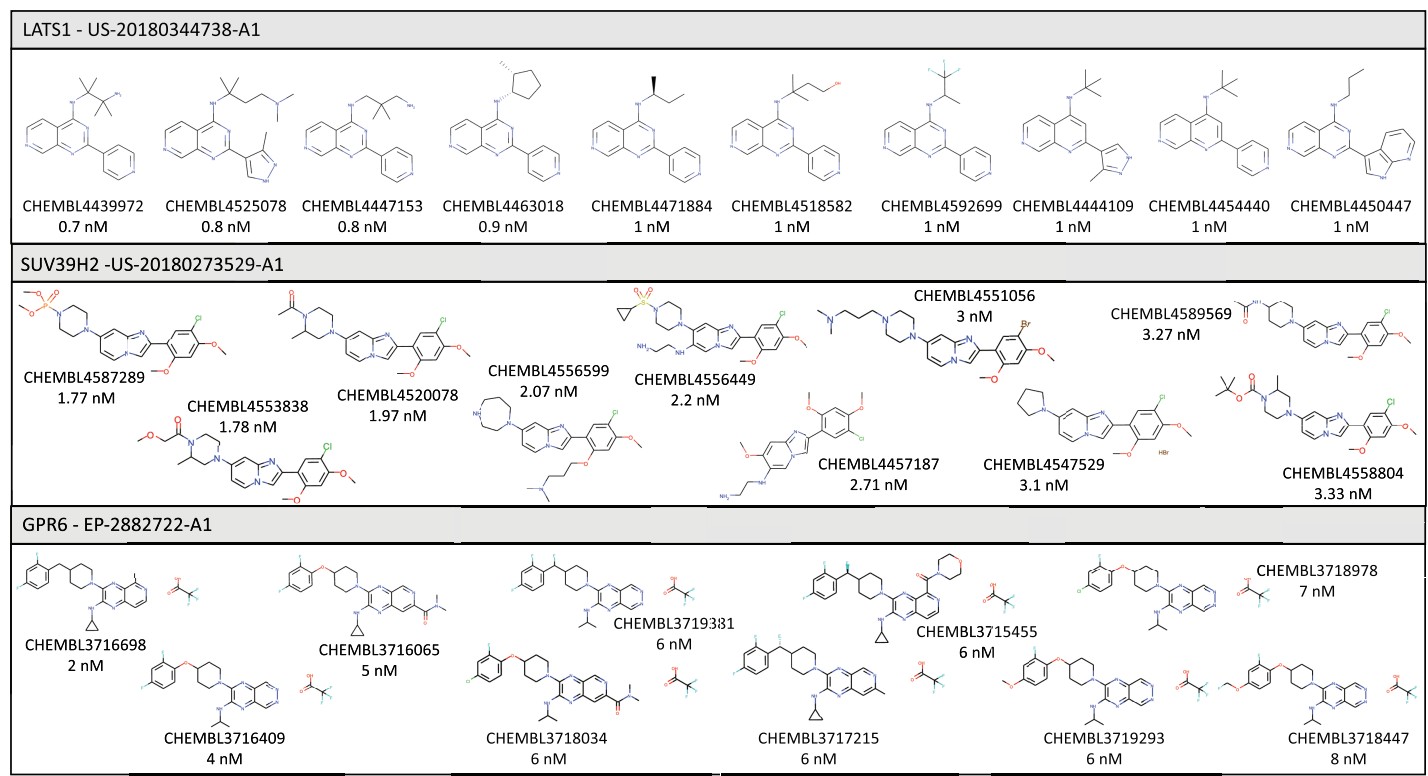

**Figure 2** Illustrative examples of bioactive compounds identified from SureChEMBL patent workflow against three targets.

It was found that the expression of this protein is elevated in some cancers, but decreased in others (*Xu et al., 2015*; *Furth & Aylon, 2017*).

Without considering data from patents, there are currently 430 molecules in ChEMBL associated with this target, extracted from 44 different articles, but only four molecules satisfy the IDG cut-off criteria for kinases. These molecules were extracted from four different articles: PMID 19654408 (*Zarrinkar et al., 2009*), PMID 22037378 (*Davis et al., 2011*), PMID 29191878 (*Klaeger et al., 2017*) and PMID 30384048 (*Narayan et al., 2019*). As a result of the current work, there are now 289 additional molecules associated with this target, with 184 molecules satisfying the IDG cut-off criteria to promote the target to the Tchem category. The source of these compounds is the patent US-20180344738-A1 (*Behnke et al., 2018*), which describes molecules designed to promote cell proliferation, with applications such as chronic wound healing, promoting liver regrowth, or treating limbal stem cell deficiency.

## Histone-lysine N-methyltransferase SUV39H2

SUV39H2 is a lysine methyltransferase, first identified in *Drosophila*, which methylates histone 3 on lysine 9 (H3K9). Di- and trimethylation of H3K9 results in gene expression repression (*Kaniskan, Martini & Jin, 2018*).

This protein is present only in embryogenesis and adult testis of healthy individuals (*Li, Zheng & Yang, 2019*), but overexpressed in several cancers, for example lung adenocarcinoma, colorectal carcinoma and gastric carcinoma (*Saha & Muntean, 2021*).

There are no selective inhibitors for this target (*Kaniskan, Martini & Jin, 2018*).

At the start of this work there were 19 molecules in ChEMBL with bioactivity data against SUV39H2, all of them from scientific literature, but none of these molecules were within the IDG cut-off.

Our patent workflow identified 460 molecules from just a single patent (US-20180273529-A1) (*Matsuo et al., 2018*), all within the corresponding IDG cut-off.

### G protein-coupled receptor 6

GPR6 is a G-protein coupled receptor, still classified as orphan by the International Union of Basic and Clinical Pharmacology (IUPHAR) (*Alexander et al., 2019*) due to lack of consistency among reports related to endogenous ligands (*Morales, Isawi & Reggio, 2018*; *Laun et al., 2019*).

It is expressed mainly in neurons in mammalian striatum and hypothalamus. There is evidence that it could have a role in several processes and diseases such as neurite outgrowth, instrumental learning, Alzheimer's disease, Parkinson's disease, Huntington's disease (*Morales, Isawi & Reggio, 2018*; *Laun et al., 2019*), and schizophrenia (*Morales, Isawi & Reggio, 2018*).

At the start of this work, there were 227 molecules in ChEMBL with bioactivity data values within the IDG cut-off. These molecules were obtained from patents, either from BindingDB, or our own curation efforts previous to the work described here.

As a result of this search, 100 additional molecules with bioactivity against GPR6 with values within the cut-off of ≤100 nM were identified, from patents WO-2018183145-A1 (*Green et al., 2018*) and EP-2882722-A1 (*Hitchcock et al., 2015*). Figure 3 shows a timeline with patent and scientific literature numbers by year for GPR6, showing that in this particular case, significantly more data were reported *via* patent disclosures than in the scientific literature. Interestingly, a new clinical candidate (currently in phase 2 clinical trials) for Parkinson's disease, CVN424 (a GPR6 inverse agonist), has been disclosed (*Sun et al., 2021*; *Brice et al., 2021*). This molecule can be found in patents as early as 2015 in patent US-9181249-B2 (*Brown et al., 2015*).

## DISCUSSION

The overall goal of this work was to identify bioactivity data on understudied targets from the patent literature, which could allow us to promote targets to the IDG Tchem category. We focused on small molecules only, but our workflow also identified several patents concerning antibodies or RNA as therapeutic agents. For the purposes of this work, we did not progress these patents further, but clearly they could also be useful in the context of "illuminating" new targets.

It should be noted that the work described here was conducted over a period of time, during which complementary data from other sources was added to the various resources concerned. This reflects the natural evolution of the underlying databases, each with their

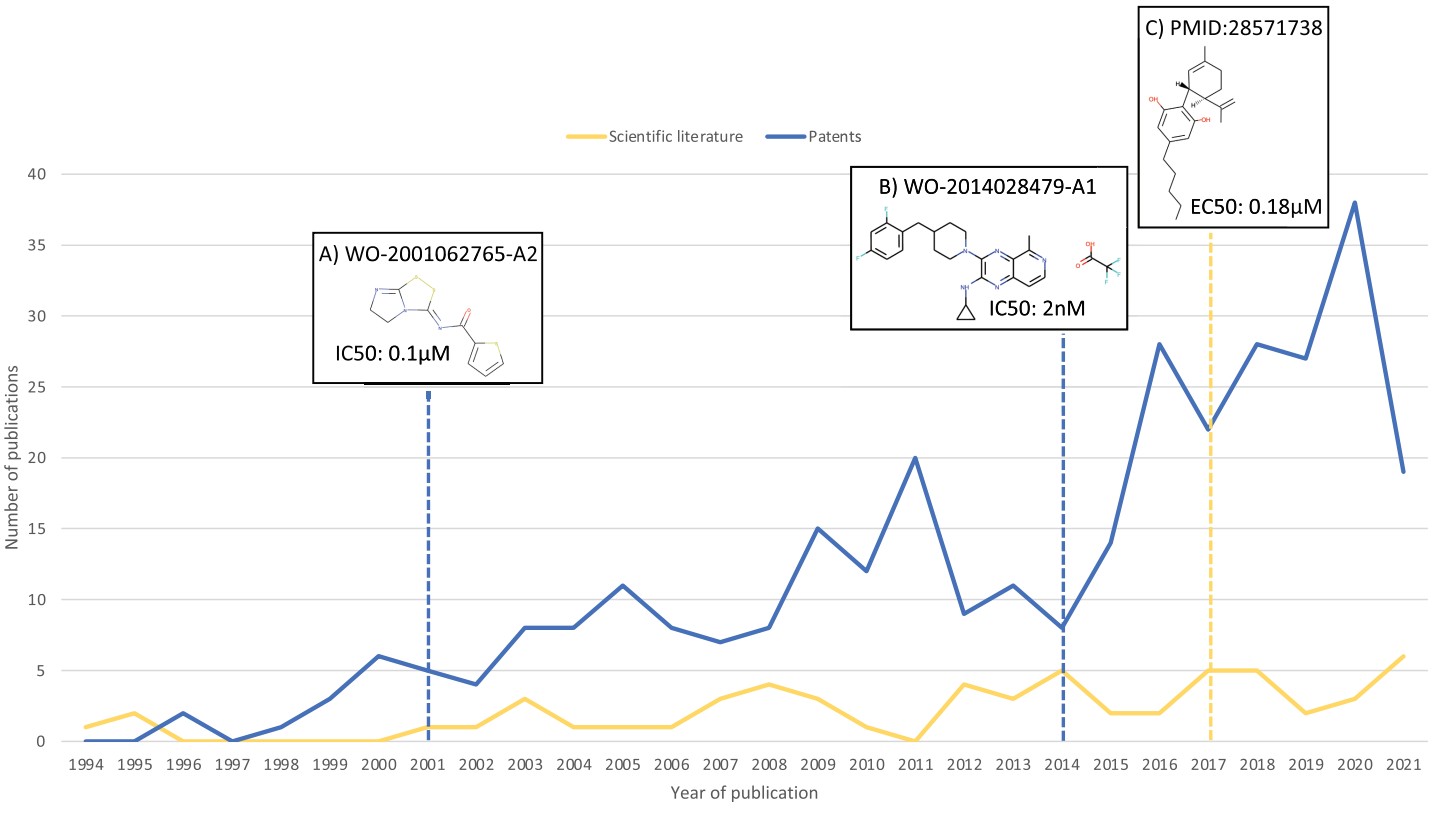

**Figure 3 Numbers of relevant patents and scientific literature publications for GPR6 per year.** Key compound disclosures in patents and scientific literature indicated by dashed lines: (A) example of a compound reported in one of the earliest patents with data against GPR6. (B) Example of a compound in one of the earliest patents identified by our method. (C) First small molecule modulator reported in scientific literature.

own update mechanism and release schedule procedures. Thus, for example, targets designated as Tdark or Tbio at the time when the research was initiated may have been promoted to a higher category based on separate evidence whilst the patent bioactivity work described here was underway. In the narrative below, the data reflects the particular snapshot corresponding to the time on which the work was initiated or completed, as appropriate.

Two of the proteins included in the list of targets that we searched for (sclerostin and exportin-1) were originally classified as Tbio at the start of this work, but later promoted to Tclin after the approval of the first-in-class drugs romosozumab and selinexor respectively.

Coincidentally, over the same period of time, bioactivity data for 30 Tbio/Tdark targets were added into ChEMBL from the scientific literature. There was an overlap of 21 targets between the two sets. This shows the value of using patents as an additional source of bioactivity data.

The work described here involved manually reviewing many patents from the six groups described in Fig. 1. As expected, the group with higher percentage of positive patents was Group 1, unexpectedly followed by the patents in Group 3, which even though they could not be flagged as containing bioactivity tables, still contained bioactivity data that was not detected automatically with the method used here, and were only found when reviewing

the patents manually. Groups 2 and 4 had lower percentages but still delivered useful and relevant patents. Even though classifying the patents in this way provided a starting point for prioritisation, clearly there are some limitations to this approach as shown in the comparison with BindingDB and the reasons for missing some patents, and for future work it would be advantageous to develop methods that can reduce the level of manual review that is required. This is the focus of currently ongoing work to develop machine-learning models able to predict which patents should be prioritised for human examination and potential curation.

A total of 74 Tdark/Tbio targets were promoted to the Tchem category on the basis of the bioactivity data identified from our patent analysis.

## CONCLUSIONS

Using relatively simple annotation and filtering pipelines, we have been able to identify a substantial number of patents containing quantitative bioactivity data for understudied targets that had not previously been reported in the peer-reviewed medicinal chemistry literature. This underlines the potential value in searching the patent corpus in addition to the more traditional peer-reviewed literature. The small molecules found in these patents, together with their measured activity against the targets, are now accessible *via* the ChEMBL database and Pharos, and have contributed to the "illumination" of previously dark targets.

## ACKNOWLEDGEMENTS

We would like to thank Dr Barbara Zdrazil for helpful comments on this manuscript.

### Funding

This work was supported by US National Institutes of Health (NIH) grants U54 CA189205 and U24 224370 (Illuminating the Druggable Genome Knowledge Management Center (IDG KMC)) at the University of New Mexico, Novo Nordisk Foundation Center for Protein Research, European Bioinformatics Institute (EBI) and University of Miami; the Wellcome Trust (Grant numbers 104104/A/14/Z and 218244/Z/19/Z); and the Member States of the European Molecular Biology Laboratory. The funders had no role in study design, data collection and analysis, decision to publish, or preparation of the manuscript.

### Grant Disclosures

The following grant information was disclosed by the authors:
US National Institutes of Health (NIH): U54 CA189205 and U24 224370.
Illuminating the Druggable Genome Knowledge Management Center (IDG KMC) at the University of New Mexico.
Novo Nordisk Foundation Center for Protein Research.
European Bioinformatics Institute (EBI) and University of Miami.

Wellcome Trust: 104104/A/14/Z and 218244/Z/19/Z.
Member States of the European Molecular Biology Laboratory.

## Competing Interests

The authors declare that they have no competing interests.

## Author Contributions

- Maria P. Magariños conceived and designed the experiments, performed the experiments, analyzed the data, prepared figures and/or tables, authored or reviewed drafts of the article, and approved the final draft.
- Anna Gaulton conceived and designed the experiments, performed the experiments, analyzed the data, prepared figures and/or tables, authored or reviewed drafts of the article, and approved the final draft.
- Eloy Félix performed the experiments, authored or reviewed drafts of the article, contributed with preparation of scripts to obtain the patents from the database, and approved the final draft.
- Tevfik Kiziloren performed the experiments, authored or reviewed drafts of the article, contributed with database queries and solving problems during database operation, and approved the final draft.
- Ricardo Arcila performed the experiments, authored or reviewed drafts of the article, contributed with database queries and solving problems during database operation, and approved the final draft.
- Tudor I. Oprea conceived and designed the experiments, authored or reviewed drafts of the article, and approved the final draft.
- Andrew R. Leach conceived and designed the experiments, analyzed the data, authored or reviewed drafts of the article, and approved the final draft.

## Patent Disclosures

The following patent dependencies were disclosed by the authors:
US-20180344738-A1 (*Behnke et al., 2018*).
US-20180273529-A1 (*Matsuo et al., 2018*).
US-9181249-B2 (*Brown et al., 2015*).
WO-2018183145-A1 (*Green et al., 2018*).
EP-2882722-A1 (*Hitchcock et al., 2015*).

## Data Availability

The list of patents and targets found in the search, the list of targets for which at least 1 compound with bioactivity data within cut-off values per target family was found, and the number of compounds found per target, are all available in the Supplemental Files.

The scripts to process the patents are available at GitHub and Zenodo: https://github.com/chembl/idg_patents_paper.

Maria Paula Magarinos & Eloy Félix. (2023). chembl/idg_patents_paper: 1.0 (1.0). Zenodo. https://doi.org/10.5281/zenodo.7669601.

## Supplemental Information

Supplemental information for this article can be found online at http://dx.doi.org/10.7717/peerj.15153#supplemental-information.

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
