# Peer review of "Illuminating the druggable genome through patent bioactivity data"

_PeerJ, doi:10.7717/peerj.15153_

## Round 0.1 · original submission · Major Revisions

We have obtained three detailed reviews of your manuscript. All reviewers are enthusiastic about your work, and they make several important suggestions for its improvement. Please address the issues highlighted by our peer reviewers. I expect this will make your manuscript stronger, more persuasive, and more impactful.

·

Basic reporting

The authors of this study report valuable data to researchers working in the field of chemical biology and/or drug discovery. The article is well written and easy to follow. One goal here was to identify novel chemicals that could target under-explored proteins through the investigation of a patent database and give access to the extracted data to the scientific community via the ChEMBL database. Figures and tables and references are appropriate.

Experimental design

Extraction and preparation of the patent data are mentioned, yet it could be of interest to the readers to add a few lines about data extraction/curation in addition to the reference Gaulton et al. 2015. This would save time to the readers. A minor point of clarification, the molecules of interest are small chemical compounds, but short peptides are also considered here ? In the discussion, the authors mention romosozumab suggesting that monoclonal antibodies are also monitored. Thus it could be valuable to clearly state Tbio, Tclin... in general (thus can include mAbs...) and the way it was used here, thus more small molecules (peptides included or not and if yes, is there a maximum number of amino acids?)

Validity of the findings

The results are clearly reported with illustrative examples. One suggestion, for readers that are eventually not familiar with ChEMBL: it might be beneficial to give more information, for example on one protein. If the readers want to investigate the results further, for instance for the LATS1 target, one would search target ID CHEMBL6167, then look at IC50...Ki values and then see the bioactivity data of the compounds and for example see compound ID CHEMBL4464290 that was not identified in a publication but from the present study (source Patent Bioactivity Data) ? Is this correct?

Additional comments

This report is of high interest for biologists and researchers working in drug discovery / chemical biology.

·

Basic reporting

1. Line 41, 68: Grammar – comma before and (for eg. selectivity, cellular activity, and potentially)
2. Line 46-49: “..than 19 million bioactivity data on almost 2 million compounds, obtained from peer-reviewed scientific literature manually extracted by curators; from other databases including PubChem BioAssay (Kim et al., 2021) and BindingDB (Gilson et al., 2016); and deposited by experimental groups.” – I would simply this statement as currently it shows discrepancy regarding whether the data from ChEMBL is via curated literature or database or both.
3. Line 78 – Reference to SureChEMBL missing
4. For the relevant studies, the authors could benefit from referencing PEMT (https://doi.org/10.1093/bioinformatics/btac716). Additionally, some of related work section seems to be 10 or more years old. I recommend do a search again to see recent developments (within past 3-5 years), if it has not yet been done.
5. Within the patent resources available, addition of Scifinder should also be cited.
6. Line 89 – I suggest simplifying the sentence as the current formation is complex for the reader to understand.
7. Line 90 – The acryonm for World Intellectual Property Organization – WIPO could be mentioned here.
8. Line 92 – The source for BindingDB’s patent data (WIPO) is missing. I recommend adding that.
9. Line 95 – The authors mention SureChEMBL is a “fully automated” database. Explaination on why it is so is missing.
10. Line 125 – A selection of databases were made for identification of Tchem targets, but no justification or reasoning on their selection is provided.
11. The quality of the figures should be improved.
12. The GitHub README could benefit from basic information on the purpose of the project alongside reference to the manuscript; Also, basic information on how to run the scripts should be added.

Experimental design

1. Was there a reason on why only years 2012-18 were searched?
2. For the IPC codes, why were A01 (related to animal husbandry), A24 (tobacco and smoking related), C05 (fertilizers), C06 (explosives), C10 (petroleum and gas) selected as a representative for life science relevant patents? It remains unclear to me on how bioactive compounds for target would be useful in the study of druggable genome.
3. Details on how was the processing done are missing. Were they done using programming scripts or some web application? Please also add the GitHub link into the main document.
4. In line 150, how was the distribution of the 62,289 patents within the six groups? Were there patents that belonged to multiple groups? Furthermore, what was the importance of classification of patent applications into the 6 groups? I did not find any evidence on the usage of the groups for further analysis.
5. Table 1 should include description on what does a positive and negative read mean.
6. Supplementary Table 1 should also include the patent count column to get a quick overview on patent applications for target classes.
7. Supplementary Table 2 should also include the uniprot accession ids for targets to be compliant with FAIR.
8. Figure 1 should also include some pre-text to allow readers to have context about the figure.
9. Line 238 – The sentence seems incomplete „.. For all the targets found, we performed a search by target..” was the search by target name or accession ids?
10. For the BindingDb comparison, a figure (probably venn diagram) providing an overview on the patents overlap would be beneficial.
11. For GPCR6, no information on pre-existing bioactivity data was reported. Please add that information as well to get an estimate of overall increase of chemicals.

Validity of the findings

In the manual curation effort to confirm the presence of selected targets, how was the annotation quality assessed? Was peer-review performed on curation meaning that multiple curator results were cross-examined to assure selection of appropriate targets. Moreover, given the no.of patent retrieved, was a crowd-sourcing of curation done?

Additional comments

The research question answered by the authors indeed provide a great example on why and how mining of patent documents could be of great advantage. It is evident from the manuscript the time complexity and work done for answering the question on how patents can actual be benefited for target data enrichment. Moreover, the comparison on patent information extracted within existing databases with the workflow described by the authors, highlights the need for generation of automated workflow or model (as the authors mentioned) to streamline and populate resources with patent data in the future. Despite the great work, I have certain concerns on the method used and they are highlighted in the sections below.

Reviewer 3 ·

Basic reporting

This is an interesting work dealing with the use of patents information (collected through SureChembl) to identify new chemobiological interactions to include in ChEMBL, especially those corresponding to underreported targets, that otherwise would be tedious and complicated to identify.

The Introduction is appropriate and provides enough background to follow the paper.

Figures are OK, but I think by including in Figure 1 the numbers and percentages for the different boxes would help to follow better the paper and the results obtained.

Experimental design

I think the research question is well defined and relevant, and the approach followed is correct.

The only issue I'd point out is that the authors do not make explicit in the Methods section that the manual review of patents was performed not on all of them, but on a subset of them. One realizes that point when one sees in Table 1 a row for "Read" patents. My question is about the selection of patents to read: was it based on a random choice? In addition, the different groups were reviewed at very different rates. How these rates where decided? I wonder if this sampling for the manual reviewing process could lead to a biased estimation of hit rates for the different groups and the whole process.

Validity of the findings

The results are interesting and show that this process can identify new chemical probes for underestudied but promising targets.

I miss however some more details and analysis of the interactions (targets + compounds) identified. Particularly the number of compounds and target classes found in the search: target class with number of compounds vs Non-IDG and IDG, etc. What (patterns) have we learned from this exercise besides finding a process to include SureChembl patents in Chembl?

On the other hand, I'm concerned about the 34 patent families in BindingDB not found in this process for the 76 targets with compounds above the activity cutoff. Since BindingDB uses a subset of the patents used by SureChEMBL, why were those 34 patent families not identified by this process, since the latter seems a quite complete one? Where did these familes end in this process? Identifying the sources of missing this data would help in improving the process.

Additional comments

At the text, the concept of "Patent FAMILY" (as opposed to just "Patent") starts to be mentioned in the Results section, but without previous definition. What do the authors mean by a patent family? This should be defined.

---

## Round 0.2 · accepted · Accept

Thank you for addressing all comments. I am glad to accept your paper for publication in PeerJ

·

Basic reporting

The authors have clarified several points in the revised version of the manuscript. It should now be ready for publication

Experimental design

This section is clear, data are provided

Validity of the findings

The results provide new insights and are valuable for the scientific community

Additional comments

The revised version of the manuscript should now be ready for publication

·

Basic reporting

no comment

Experimental design

no comment

Validity of the findings

no comment

Additional comments

All my comments have been resolved.

Reviewer 3 ·

Basic reporting

The reviewers have addressed my concerns and I think the work is now publishable

Experimental design

The reviewers have addressed my concerns and I think the work is now publishable

Validity of the findings

The reviewers have addressed my concerns and I think the work is now publishable